# Effects of physical activity and exercise on well-being in the context of the Covid-19 pandemic

**Juliana Marques de Abreu**[1], **Roberta Andrade de Souza**[1], **Livia Gomes Viana-Meireles**[1], **J. Landeira-Fernandez**[2], **Alberto Filgueiras**[3]*

1 Instituto de Educação Física e Esportes, Universidade Federal do Ceara, Fortaleza, Brazil,
2 Departamento de Psicologia, Pontifícia Universidade Católica do Rio de Janeiro, Rio de Janeiro, Brazil,
3 Departamento de Cognição e Desenvolvimento, Universidade do Estado do Rio de Janeiro, Rio de Janeiro, Brazil

* albertofilgueiras@gmail.com

**Data Availability Statement:** All relevant data are within the manuscript and its Supporting Information files.

## Abstract

Coronavirus disease 2019 (COVID-19) was discovered in China and characterized by the World Health Organization as a pandemic in March 2020. Many countries worldwide implemented stringent social isolation as a strategy to contain virus transmission. However, the same physical distancing that protects against the spread of COVID-19 may negatively impact mental health and well-being of the population. The present study sought to shed light on this phenomenon by assessing the relationship between physical activity and subjective well-being (SWB) among individuals who were subjected to social isolation during the COVID-19 pandemic. Data were collected in Brazil between March 31 and April 2, 2020. All of the volunteers agreed to participate by digitally checking the option of agreement after reading consent terms. The inclusion criteria were participants who had been in social isolation for at least 1 week and agreed to the consent terms. Three instruments were applied. A questionnaire was constructed for this study that assessed the participants' exercise routines. The Psychosocial Aspects, Well-being, and Exercise in Confinement (PAWEC) scale was created by researchers of this study that assessed the relationship between well-being and physical activity during social isolation. The Brazilian Portuguese-adapted version of the Positive and Negative Affect Schedule (PANAS) was also used. A total of 592 participants (371 female, 220 male, 1 transgender), 14–74 years old (M = 32.39 years, SD = 10.5 years), reported being in social isolation for an average of 14.4 days (SD = 3.3 days). Well-being that was related to the practice of physical activity during quarantine was linked to an established routine of physical activity before the social isolation period. Participants who already practiced physical exercises previously and reported continuing the practice during the quarantine period had higher positive affect scores. Participants who engaged in physical activity without direct guidance only during the quarantine period had higher negative affect scores. Participants who already practiced physical activity felt more motivated to continue practicing physical activity during the social isolation period, resulting in positive affect, unlike participants who began exercising only during quarantine. Our results suggest that negative affect can occur among individuals who only just begin exercising during social

**Funding:** This study is funded by Grant #2020.09837.43 from the Program Cientista do Nosso Estado of FAPERJ (Fundação Carlos Chagas de Apoio à Pesquisa do Estado do Rio de Janeiro) and Grant Produtividade Pq1A of the CNPq (Conselho Nacional de Desenvolvimento Científico e Tecnológico) both awarded to JLF. The funders had no role in study design, data collection and analysis, decision to publish, or preparation of the manuscript. The funders had no role in study design, data collection and analysis, decision to publish, or preparation of the manuscript. The authors receive no salary from funders.

**Competing interests:** The authors have declared that no competing interests exist.

isolation, indicating that physical activity should be habitual and not only occur during periods of social isolation. Engaging in exercise only during social isolation may contribute to an increase in malaise.

## Introduction

Coronavirus disease 2019 (COVID-19) was discovered in Wuhan, China, in December 2019. It was soon declared an international public health emergency [1]. COVID-19 is an infectious disease that is caused by severe acute respiratory syndrome-coronavirus 2 (SARS-CoV-2) respiratory infection. On March 11, 2020, COVID-19 was characterized by the World Health Organization (WHO) as a pandemic. One of the preventive measures that was suggested by the WHO to contain contagious spread was physical distancing between people, instructing everyone to stay in their homes [2–5]. The interruption of daily activities, such as work, studies, and leisure, among others, occurred in an attempt to prevent an increase in cases. Since the beginning of March 2020, most states in Brazil adhered to social isolation and quarantine [6]. Social isolation occurs when a person or group of people either voluntarily or involuntarily withdraw from social interactions and activities to lower the chances of the spread of disease [7].

The Ministry of Health in Brazil stated on March 11, 2020, "Isolation policies aim to separate symptomatic and asymptomatic people diagnosed with COVID-19 from the rest of the population in order to prevent the spread of infection and local transmission" [6]. However, physical distancing that prevents the spread of COVID-19 may negatively impact mental health and well-being of the population. A 2004 study of 129 Canadians who were quarantined during the SARS epidemic found symptoms of posttraumatic and depressive stress. These symptoms were directly related to age, education level, living with other adults, not having children, and the quarantine duration. A longer time in isolation was associated with a greater risk of symptoms [7]. Similar results were found in studies of SARS-related quarantine in Taiwan [8] and Hong Kong [9]. Filgueiras and Stults-Kolehmainen [10] recently investigated psychosocial factors among Brazilians in quarantine during the COVID-19 pandemic and found that gender, quality of nutrition, attendance in tele-psychotherapy, exercise frequency, the presence of older adults in quarantine with the person, obligation to work outside, education level (higher education was associated with a lower risk of mental illness), and age (younger age was associated with a higher risk of mental illness) predicted depression and anxiety states.

According to an empirically tested definition, subjective well-being (SWB) refers to understanding how people assess their own lives. Such assessments must be cognitive (i.e., overall satisfaction with life and other specific domains, such as marriage and work) and include a personal analysis of the frequency with which positive and negative emotions are experienced. To achieve an adequate level of SWB, the individual should recognize higher levels of life satisfaction, a high frequency of positive emotional experiences, and a low frequency of negative emotions [11]. This leads to the assumption that lower levels of SWB are linked to higher levels of psychosocial symptoms, such as anxiety, depression, and stress [12]. Evidence supports this hypothesis in different populations, including children and adolescents [13], young adults [12], and older adults [14].

Recent evidence supports the pivotal role of physical activity and exercise in lowering stress, depression, and anxiety [15–19]. The International Society of Sport Psychology [20] published a consensus statement that linked physical activity and psychological benefits. Their

conclusion was that long-term exercise is generally associated with lower anxiety, stress, and depression and an increase in self-esteem and positive emotions. Physical activity can also be beneficial for the immune system in its fight against COVID-19 [21].

According to a previous meta-analysis [16], sessions of 20–60 min, 3–5 times weekly, with an intensity between 60% and 90% of the maximum cardiac frequency are key factors in the ability of physical exercise to generate more consistent psychological benefits [22–25]. Thus, physical activity is hypothesized to decrease psychosocial symptoms and negative affect and increase positive affect and overall SWB. However, unclear is whether people in quarantine are able to engage in such a frequency and regularity of exercise to increase SWB. The present study sought to shed light on this phenomenon by assessing the relationship between physical activity and SWB among individuals in social isolation during the COVID-19 pandemic.

# Methods

## Sample

The sample consisted of 592 participants (371 women [62.7%], 220 men [37.2%], 1 transgender [0.01%]), with a mean age of 32.3 years (SD = 10.5 years). All of the volunteers agreed to participate by digitally checking the option of agreement after reading the consent terms.

## Procedures

This study was conducted between March 31 and April 2, 2020. Data collection was performed during the first 15 days of quarantine because we wanted to capture subjects' initial feelings about the lockdown. The Ethical Committee of Pontifical Catholic University of Rio de Janeiro, Brazil (approval no. 2020.876–459), approved this research for online data collection. All of the procedures were in accordance with the Declaration of Helsinki and ethical guidelines of Brazilian authorities. Volunteers were recruited via social media and smartphone messaging applications. Upon receiving the link to answer the questionnaire, the participants had access to a research presentation and the consent terms that explained that this study was voluntary and not mandatory and that the information obtained would be kept anonymous. After agreeing to participate, individuals who did not accept the consent terms were directed to a "Thank you" page. Individuals who accepted the consent terms were directed to a demographic questionnaire. The same order of questions and instruments were applied to all of the participants: (1) consent terms, (2) demographic questionnaire, (3) exercise routine questionnaire, (4) PAWEC, (5) PANAS, and (6) "Thank you" page. All of the volunteers responded to all items of the instruments.

The inclusion criteria were participants over 18 years old who had been in social isolation for at least 1 week and agreed to the consent terms. The exclusion criteria were volunteers with a history of any kind of psychiatric condition, even those under treatment, and who self-reported to be sedentary.

## Instruments

Three instruments were administered online and sent via a single form via Google Docs. Respondents first had access to the consent terms. After agreeing to participate, the sociodemographic section of the demographic questionnaire was presented, which asked about age, gender, education, number of days in quarantine, physical activity, and exercise habits, followed by presentation of the exercise routine questionnaire, PAWEC, and PANAS in separate sections.

The exercise routine questionnaire was created specifically for this study. It was an 8-item instrument that assessed the participants' exercise routines (e.g., "Have you been monitored by an online fitness coach during quarantine?" "Were you monitored by a fitness coach before

isolation?" "Did you use any media source [e.g., YouTube, social media, videos, smartphone apps, etc.] to exercise before the quarantine?"). The participants answered "Yes" or "No." One open-ended question asked the respondent to indicate "the number of days you practiced exercise or physical activity before isolation."

The PAWEC was also created by researchers of the present study. It assessed the relationship between well-being and physical activity during the social isolation period. It is an 18-item questionnaire which assesses whether the frequency of physical activity and exercise has a positive or negative influence on psychological aspects and SWB, namely mood, happiness, motivation, anxiety, and sadness (e.g., Item 7: "Do you feel happy while exercising in quarantine?" Item 9: "How often do you believe it is important to exercise during isolation?" Item 13: "Do you feel anxious whenever you exercise during quarantine?").

The Brazilian Portuguese-adapted version of the PANAS was developed by Pires et al. [26]. This 20-item instrument was originally developed by Watson, Clark, and Tellegen in1988. It measures Positive Affect (PA) and Negative Affect (NA), defined as general dimensions that describe the affective experience of individuals. High NA scores reflect subjective displeasure and malaise, including such emotions as fear, nervousness, and disturbance. High PA levels reflect subjective pleasure and well-being, including such emotions as enthusiasm, inspiration, and determination. The Brazilian version of the PANAS has a two-factor structure with a significantly moderate negative correlation between factors ($r$ = -0.42) and reliability (Cronbach's $\alpha$ = 0.84, ranging from $\alpha$ = 0.88 in for PA to $\alpha$ = 0.90 for NA) [26].

## Data analyses

Descriptive statistics were calculated according to the nature of the measure. Frequencies and percentages are presented for categorical data. Arithmetic means and standard deviations (SDs) are presented for continuous data. Cronbach's $\alpha$ was computed to investigate preliminary reliability of the PAWEC. Preliminary validity was assessed using Exploratory Factor Analysis (EFA), which was performed by adopting the recommendations of [26] for ordinal variables. Parallel Analysis was used to determine the number of factors using a polychoric correlation matrix. Unweighted Least Square factor modeling was performed to assess factor retention. Direct Oblimin rotation was adopted as an oblique method when needed because of the expectancy of correlated factors, although significant negative correlations were expected.

After ensuring sufficient reliability and validity of both measures that were developed for this study, linear multiple regression (LMR) was separately computed for PA and NA scores on the PANAS as dependent variables. The stepwise method was adopted for these regressions. The first step of the regression was the PAWEC total score. Demographic variables (i.e., age, gender, education, and number of days in quarantine) and exercise routine variables were considered to predict the results of both PANAS factors. The second step of the regression comprised PAWEC items independently. The significance level for variable inclusion in the LMR was $p < 0.05$. The $\beta$ coefficient revealed the strength of the association between independent and predicted variables. Additionally, $t$-test statistics (in addition to $p$ values and effect sizes) were computed to assess whether one variable would or would not be included in the LMR. Analysis of variance (ANOVA) was used to compare the LMR model to the null hypothesis (i.e., the constant). The effect size was measured by $f^2$ statistics of respective ANOVAs by considering the following interpretation: > 0.02 and < 0.15 were considered a small effect size, > 0.15 and < 0.35 were considered a moderate effect size, and > 0.35 was considered a large effect size. Descriptive statistics and LMR were computed using R software with the psych package. Exploratory Factor Analysis was performed using FACTOR software [27]. Effect sizes were calculated using G*Power 3.1.9.2 software.

## Results

Demographics characteristics of the participants are shown in "Table 1".

The data showed that the participants engaged in physical activities an average of 4.5 days (SD = 1.2) per week. Categorical variables of exercise routines are shown in "Table 2".

The frequencies of types of exercise changed before and during quarantine, although we could not infer statistical differences. "Table 3" shows a comparison of exercise before and during social isolation.

The psychometric properties, assessed by the PAWEC, were provided before proceeding with the other analyses. The preliminary EFA results showed sample adequacy for $KMO = 0.849$ and a significant Bartlett sphericity test: $\chi^2 = 3509.238$, $df = 153$, $p < 0.001$. The parallel analysis revealed that the best solution was a three-factor structure that cumulatively explained 53.54% of the variance. "Table 4" shows factor loadings of items and percentages of explained variance per factor.

Three dimensions were then named based on the content of items that loaded on the same factor: exercise effects (positive or negative affect associated with exercise during quarantine), cognition (cognitive variables that entail understanding the reasons for and importance of exercise during quarantine), and mood (motivational and emotional aspects that are involved in exercise during quarantine). The reliability of the entire PAWEC was $\alpha = 0.84$ (exercise effects: $\alpha = 0.77$, cognition: $\alpha = 0.71$, mood: $\alpha = 0.82$).

After ensuring validity and reliability of the PAWEC, the first LMR that was computed was the PA score of the PANAS as the dependent variable. The model was significant ($F_{5,30} = 30.850$, $p < 0.001$, $f^2 = 0.06$), with 19% of the variance explained according to $r^2 = 0.19$. For the NA factor of the PANAS, the model was also significant ($F_{7,28} = 35.498$, $p < 0.001$, $f^2 = 0.08$), with a coefficient of determination of $r^2 = 0.22$, suggesting that 22% of the variance of NA was explained by the independent variables.

With regard to the use of media resources to assist in the practice of physical activity, 463 participants (75.3%) reported that they did not use them before the beginning of social isolation, whereas 152 (24.7%) stated they did. During social isolation, this number changed significantly to 365 (59.3%) who used media resources and 250 (40.7%) who did not. A total of 515 participants (83.7%) reported that they were able to perform physical activities without professional monitoring, whereas 100 (16.3%) reported they were not. A total of 257 participants

**Table 1. Demographics characteristics of the participants.**

|  | Mean | No. (%) |
|---|---|---|
| Days in isolation | 14.4 (SD = 3.3) | |
| Number of people living with you during quarantine | 2.7 (SD = 2.3) | |
| Physical activities (days per week) | 4.5 (SD = 1.2) | |
| Level of education | | |
| | Completed elementary school | 150 (25.3%) |
| | High school degree | 172 (28.9%) |
| | College | 28 (4.7%) |
| | Bachelor's degree | 41 (8.1%) |
| | Master's degree | 35 (5.9%) |
| | PhD degree | 343 (57.9%) |
| Marital status | | |
| | Single | 343 (57.9%) |
| | Married | 189 (31.9%) |

**Table 2. Descriptive statistics of categorical variables of exercise routines.**

| Question | Frequency | |
|---|---|---|
| | No. | % |
| Were you monitored by a fitness coach before quarantine? | | |
| Yes | 495 | 83.60% |
| No | 97 | 16.40% |
| Have you been monitored by an online fitness coach during quarantine? | | |
| Yes | 252 | 42.60% |
| No | 340 | 57.40% |
| Did you use any media source before quarantine? | | |
| Yes | 143 | 24.20% |
| No | 449 | 75.80% |
| Did you use any media source during quarantine? | | |
| Yes | 352 | 59.50% |
| No | 240 | 40.50% |
| Are you practicing exercise more frequently during the quarantine? | | |
| Yes | 101 | 17.10% |
| No | 491 | 82.90% |

(41.8%) reported that they received professional guidance about physical activities during the pandemic, whereas 358 (58.2%) did not.

Positive affect entails pleasant feelings or emotions, such as joy, happiness and enthusiasm. The PAWEC added a positive affect dimension to assess participants' affect related to exercise. The frequency of physical exercise before social isolation and less use of social medias were associated with higher a positive influence. Negative affect comprises negative feelings and emotions such as sadness, laziness, nervousness and irritability. The PAWEC adopted a negative affect dimension to measure how bad participants feel whenever they do exercise during the quarantine. These dimensions were influenced by the number of days in quarantine (i.e., more days in quarantine was associated with a higher negative influence), frequency of exercise during isolation and gender.

## Discussion

The present study sought to understand the relationship between physical activity and SWB among individuals who were subjected to social isolation at the beginning of the COVID-19

**Table 3. Type of physical exercise before and during social isolation.**

| Physical exercise | Before | | During | |
|---|---|---|---|---|
| | No. | % | No. | % |
| Strength training | 31 | 5.2% | 82 | 13.9% |
| Functional training | 26 | 4.4% | 292 | 49.3% |
| Yoga/Pilates | 63 | 10.6% | 36 | 6.1% |
| Martial arts/fighting | 107 | 18.1% | 10 | 1.7% |
| Walking/running | 27 | 4.6% | 71 | 12.0% |
| Dance/zumba | 28 | 4.7% | 27 | 4.6% |
| Bicycle training | 162 | 27.4% | 19 | 3.2% |
| Swimming | 99 | 16.7% | 2 | 0.3% |
| Other type of exercise | 49 | 8.3% | 53 | 9.0% |

**Table 4. Exploratory factor analysis of the PAWEC.**

| Item | Factor loading | | |
|---|---|---|---|
| | Factor 1 | Factor 2 | Factor 3 |
| 1. Do you feel good by practicing exercise during quarantine? | 0.93 | -0.08 | -0.04 |
| 7. Do you feel happy by practicing exercise during quarantine? | 0.91 | -0.06 | -0.04 |
| 6. Do you feel more energetic the day you practice exercise, during quarantine? | 0.84 | 0.06 | 0.02 |
| 2. Do you feel well on days you practice exercise during quarantine? | 0.81 | -0.04 | 0.06 |
| 10. Do you feel bad whenever you do not practice exercise during quarantine? | -0.51 | -0.14 | 0.16 |
| 16. Do you feel less energetic whenever you do not practice exercise during quarantine? | -0.40 | 0.03 | 0.19 |
| 9. Do you believe it is important to exercise during quarantine? | 0.03 | 0.67 | 0.27 |
| 15. Do you believe it is necessary to exercise during quarantine? | 0.04 | 0.66 | 0.29 |
| 18. Do you believe that exercise during quarantine is unnecessary? | -0.11 | -0.51 | -0.13 |
| 4. Do you feel your day gets better whenever you practice exercise during quarantine? | 0.03 | 0.45 | 0.10 |
| 11. Do you feel your day gets worse whenever you practice exercise during quarantine? | 0.08 | -0.35 | -0.14 |
| 8. Do you feel more motivated to exercise during quarantine? | 0.12 | 0.24 | 0.53 |
| 3. Does your mood improve whenever you practice exercise during quarantine? | 0.02 | 0.09 | 0.51 |
| 17. Do you feel unmotivated to exercise during quarantine? | -0.14 | 0.03 | -0.47 |
| 5. Do you feel less anxious while practicing exercise during quarantine? | 0.25 | 0.11 | 0.43 |
| 14. Do you feel more anxious on days you do not practice exercise during quarantine? | -0.02 | -0.16 | -0.42 |
| 13. Do you feel sad whenever you do not practice exercise during quarantine? | -0.11 | -0.04 | -0.37 |
| 12. Does your mood worsen whenever you do not exercise during quarantine? | -0.22 | 0.03 | -0.31 |
| Explained variance | 23.99% | 17.13% | 14.42% |

pandemic. Engaging in physical activity during the initial phase of the pandemic was associated with greater SWB. In fact, emotional effects of exercise were predicted by frequency of exercise, gender, number of days in social isolation, the use of media resources and physical activity routine before quarantine.

The main goal of this study was to understand the relationship between aspects of exercise and SWB. We developed an instrument to measure personal attitudes about exercise during quarantine. The PAWEC was created for this study. The EFA indicated that a three-factor model best explained latent structures of physical activity among individuals in social isolation. We conducted the EFA according to Lorenzo-Seva [28], which revealed good sample adequacy and capacity for factorial rotation, which was required because the instrument showed a multi-dimensional structure. The first dimension of the PAWEC was mood (i.e., either a positive or negative influence of exercise among people in quarantine). The second dimension of the PAWEC that was identified by the EFA was exercise effects (i.e., motivational and emotional positive aspects involved in exercising during social isolation, such as feeling happy or feeling more energetic after exercise). The third dimension of the PAWEC was cognition (i.e., understanding reasons and importance of exercise during confinement, such as believing the necessity of exercise and understanding the importance of physical activity). All three factors separately and PAWEC as a whole had Cronbach's $\alpha > 0.70$, which was considered sufficient to assume good internal consistency. Thus, the PAWEC was both reliable and well structured.

## Exercise effects on positive and negative affection

The results suggested that the exercise effects dimension of the PAWEC correlated with both NA and PA on the PANAS. A person who does not enjoy exercising will probably not feel either positive or negative aspects of SWB. Neurophysiological and social benefits of regular

physical activity are well known [29–31]. These benefits are also maintained during social isolation, such as during the COVID-19 pandemic, especially when we consider the mood factor. However, positive outcomes of SWB rely on subjective feelings of happiness and energy after exercising, whereas poorer SWB was associated to the negative effects of exercising. For individuals who do not feel any changes after exercising, their SWB will remain unaltered.

Another dimension of the PAWEC, exercise effects, was also significantly related to both positive and negative aspects of SWB. Previous studies showed that exercise leads to improvements in well-being and mental health outcomes [16, 17, 25]. Accordingly, feeling better during COVID-19-related quarantine can be achieved by engaging in an exercise routine, which can prevent increases in depression, anxiety, and stress [10]. These findings corroborate previous studies that suggested that frequent exercise can help achieve positive aspects of SWB.

Some studies highlighted the relevance of an individual's awareness of the beneficial effects of exercise to improve SWB [32]. The present study suggests that awareness of the role of exercise does not necessarily lead to either an increase or decrease in SWB among people who are quarantined during the COVID-19 pandemic. Thus, simply stating that exercising during quarantine is important does not influence people's SWB. Exercise routines should be prepared that are joyful and playful, rather than being concerned about explaining the relevance or importance of exercise, at least in situations of social isolation.

### Influence of gender, number of days of quarantine, type of exercise, and use of social media on subjective well-being

Beyond participants' attitudes about exercise, we also collected more objective variables. Gender correlated with negative aspects of SWB. Because of the coding that we employed (i.e., men were coded 1, women were coded 2), the positive linear relationship between NA and gender means that women had higher NA compared with men. This result was supported by previous findings that women suffer more anxiety, stress, and depression than men during COVID-19-related quarantine [31], indicating that women feel more discomfort during such situations.

We also found an association between the number of days in quarantine and NA. A longer quarantine period was associated with worse feelings about SWB. This finding corroborates evidence from other studies of mental health among people who were subjected to epidemic-related quarantine [7–9]. A long time in social isolation and confinement increases the risk for mental health symptoms. Thus, quarantine must be a temporary rather than a permanent measure for avoiding possible virus spread.

Participants answered questions about their frequency and type of physical activity. Although the type of exercise did not differ in the present study, two exercise-related variables were significantly associated with SWB: frequency of exercise before quarantine and an increase in the frequency of exercise during quarantine. The literature indicates that 3–5 days of moderate exercise weekly increases PA and decreases NA aspects of SWB [16, 23]. Participants who reported that they exercised before the quarantine and kept exercising during the quarantine had a higher association with positive aspects of SWB than participants who did not engage in physical activity routines before the quarantine. Increasing the frequency of exercise was positively associated with negative aspects of SWB, meaning that participants who reported exercising more during the quarantine felt worse than those who maintained their usual exercise routines. Stuart and Nanette [16] suggested that a sudden increase in exercise and exhaustive exercises more than 5 days per week may lead to lower SWB. Our findings suggest that achieving a proper balance is pivotal for SWB. Both being sedentary and engaging in exaggerated exercise routines may increase NA. Thus, physical activity programs that are

introduced to achieve a beneficial effect must be individualized and likely developed by physical education professionals who are knowledgeable about training principles to achieve maximum exercise performance and protect against harmful excess, including injury [33]. Thus, individuals who engaged in physical activity only after the quarantine was imposed may not have taken these principles into account, which can generate feelings of discomfort.

Filgueiras and Stults-Kolehmainen [10] investigated mental health during COVID-19-related quarantine and found that sudden changes in habits during the isolation period increased the risk of depression, anxiety, and acute stress. This result corroborates the present findings. Quarantine makes people have a sense that starting regular physical activity could contribute to improvements in physical and mental health. However, the sudden inclusion of an exercise routine, often without prior planning, may contribute to a greater sense of subjective malaise.

More frequent exercise is associated with a lower incidence of psychological disorders. Many stressors tend to undergo a reduction of strength when exercise habits increase [34]. The inclusion of physical activity only during the quarantine period was not associated with greater SWB. This result suggests that the inclusion of healthier habits during the social isolation period should not begin solely with an increase in physical activity.

The sudden introduction of physical activity and lack of individualized planning can lead to another issue that was identified in the present study, namely the relationship between SWB and the adoption of social media and other internet resources to build exercise routines. Participants who used technological yet impersonal applications (i.e., YouTube, Instagram, and other mobile applications) to conduct physical activity exhibited a decrease in PA and an increase in NA. Such online training design programs do not consider the reality of each individual. Instead, they provide only general guidance and thus only meet the needs of specific groups of people [33]. Although it could be expected that such resources are beneficial, they can actually decrease PA and increase NA and thus are not recommended for people in COVID-19-related quarantine.

Based on the present results, physical activity during quarantine can contribute to greater SWB for people during quarantine but only for individuals who already engaged in habitual exercise before quarantine. Including a new exercise habit only during quarantine, which requires guidance and care during isolation, can increase NA. Additionally, the indiscriminate use of media resources to initiate physical activity can elicit a greater sense of discomfort. The practice of physical activity during periods of social isolation should be designed to be closer to prior habits. Feeling obligated to exercise during social isolation when this was not previously a reality for the person can increase malaise. Other strategies, such as having a balanced diet and regular eating habits [10] and getting involved in artistic activities, may be better ways to increase SWB.

## Conclusion

The present study found that the practice of physical exercise during COVID-19-related social isolation affected both positive and negative aspects of SWB. People who engaged in physical exercise during quarantine when they did not previously do so could have resulted in feeling "pressured" to spend time engaged in physical exercise, and such individuals did not always receive professional guidance in their training. When not planned, targeted, or individualized, workouts can lead to feelings of discomfort and NA. However, people who practiced physical exercise before quarantine and continued to do so (e.g., practicing more than twice weekly) had relatively good SWB, thus maintaining the beneficial effects of exercise in both non-confinement and confinement contexts.

Participants who engaged in light exercise also exhibited an increase in SWB and satisfaction with oneself. Practicing physical exercise during quarantine may be considered positive from the perspective of SWB, as long as there is no need for drastic changes in attitudes or prior habits. Physical exercise among people who did not practice previously but rather decided to begin such practices during quarantine and without proper guidance from a professional did not generate positive emotional effects.

One limitation of the present study was the period of data collection, which occurred only at the beginning of the pandemic and outset of quarantine. Nonetheless, the present study highlights the importance of physical education professionals to devise individualized training routines. Another contribution of this study was generation of the PAWEC, which can be used to study other isolation contexts, such as prison confinement or the isolation of workers on offshore oil and gas platforms where individuals are isolated from most of society.

## Supporting information

**S1 Data.**
(XLSX)

## Acknowledgments

The authors acknowledge support from Coordenação de Pessoal de Nível Superior, Brazil.

## Author Contributions

**Conceptualization:** Juliana Marques de Abreu, Roberta Andrade de Souza, Livia Gomes Viana-Meireles, Alberto Filgueiras.

**Data curation:** Juliana Marques de Abreu, Roberta Andrade de Souza, Livia Gomes Viana-Meireles, Alberto Filgueiras.

**Formal analysis:** Alberto Filgueiras.

**Funding acquisition:** J. Landeira-Fernandez.

**Investigation:** Juliana Marques de Abreu, Roberta Andrade de Souza, Livia Gomes Viana-Meireles, J. Landeira-Fernandez.

**Methodology:** J. Landeira-Fernandez, Alberto Filgueiras.

**Project administration:** J. Landeira-Fernandez.

**Supervision:** Livia Gomes Viana-Meireles.

**Writing – original draft:** Juliana Marques de Abreu, Roberta Andrade de Souza, Livia Gomes Viana-Meireles, J. Landeira-Fernandez, Alberto Filgueiras.

**Writing – review & editing:** Livia Gomes Viana-Meireles, J. Landeira-Fernandez, Alberto Filgueiras.

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
