## [Decision Letter · Decision Letter 0]

22 Jan 2021

PONE-D-20-25501

Effects of physical activity and exercise on well-being in the context of the Covid-19 pandemic

PLOS ONE

Dear Dr. Filgueiras,

Thank you for submitting your manuscript to PLOS ONE. After careful consideration, we feel that it has merit but does not fully meet PLOS ONE’s publication criteria as it currently stands. Therefore, we invite you to submit a revised version of the manuscript that addresses the points raised during the review process.

Whereas all three reviewers highlight that the manuscript covers an important topic and uses an interesting approach, they however also raise a number of substantial concerns regarding the clarity of the writing and the information around methodological and data acquisition and analyses issues. If you decide to resubmit a revised version you should carefully consider all the reviewers’ suggestions.

We look forward to receiving your revised manuscript.

Kind regards,

Anke Karl

Academic Editor

PLOS ONE

Journal Requirements:

2. Thank you for stating the following in the Acknowledgments and the Founding Source Sections of your manuscript:

"Authors thank the support of the

Coordenacao de Aperfeicoamento de Pessoal de Nivel Superior (CAPES) and the Fundacao

Carlos Chagas de Amparo a Pesquisa do Estado do Rio de Janeiro (FAPERJ) to conduct this

study."

"The National Council for Scientific and Technological Development under Produtividade

PQ-1A Grant from the author JLF funded this study."

Reviewers' comments:

Reviewer's Responses to Questions

**Comments to the Author**

1. Is the manuscript technically sound, and do the data support the conclusions?

Reviewer #1: Yes

Reviewer #2: Yes

Reviewer #3: Partly

2. Has the statistical analysis been performed appropriately and rigorously? 

Reviewer #1: Yes

Reviewer #2: Yes

Reviewer #3: I Don't Know

3. Have the authors made all data underlying the findings in their manuscript fully available?

Reviewer #1: No

Reviewer #2: Yes

Reviewer #3: Yes

4. Is the manuscript presented in an intelligible fashion and written in standard English?

Reviewer #1: Yes

Reviewer #2: Yes

Reviewer #3: Yes

5. Review Comments to the Author

Reviewer #1: The manuscript covers a very interesting topic and the approach used by the authors is quite robust. However, the way it is written and structured makes it hard for the reader to follow. In addition, some information needs to be better explained by the authors. Detailed comments:

- The manuscript would be both more compelling and useful to a broad readership if the authors moved beyond providing a descriptive narrative of the data analyzed in an assembled way and be more forthright and clear in presenting the instruments and the results (maybe using tables).

- The number of acronyms should be considerably reduced, as it makes the text hard to follow, particularly in the methods section. By the way, the acronym of Positive Affect should be PA and not AP.

- It is not clear if all participants answered all the questionnaires’ items.

- The way results are presented is rather confusing: subtopics should be added to direct the reader.

- The correlation between the sociodemographic data collected and the instruments should be clearer and better explored.

- A "Limitations" section at the end of the "Discussion" and as well as a “Conclusion” section should be included. In the “Conclusion” section, policy implications of the results achieved should be added.

Reviewer #2: The manuscript is of general interest and conducted with a rigorous research approach. I am satisfied with the work done by the authors. I have only the following minor comments to rise.

ABSTRACT

- The authors reported the acronym SWB without having defined it first.

- Please provide age of the sample and gender distribution.

- Why did the authors only report strength training results? Authors should better report the main findings found.

INTRODUCTION

- The introduction is too long and dispersed. Authors should streamline it and make it converge on the topics related to the study (for example, the description of the symptoms of COVID-19 or the difference between symptomatic and asymptomatic is not necessary for the purpose of the study).

- Some sentences are not supported by bibliographic sources (for example the following sentence: “However, the same physical distance, which protects COVID-19 from spreading, may have an impact on mental health and the well-being of the population.” For the latter statement I suggest considering the following references:

1) Giustino, V. et al. Physical Activity Levels and Related Energy Expenditure during COVID-19 Quarantine among the Sicilian Active Population: A Cross-Sectional Online Survey Study. Sustainability 2020, 12, 4356.

2) López-Bueno, R. et al. Health-Related Behaviors Among School-Aged Children and Adolescents During the Spanish Covid-19 Confinement. Frontiers in paediatrics 2020, 8, 573.

DISCUSSION

- In the discussion, same sentences concerning the PAWEC should be reported in the methods section (for example the following sentence: “the scale as a whole showed Cronbach’s alpha above 0.70, which was considered enough to assume good internal consistency. The instrument, then, proved to be reliable and well structured”).

- Before describing the results found, the authors should report what their hypothesis was.

- The authors failed to report strength and limitations of the study. Among the latter, the self-reported based instrument should be mentioned.

Reviewer #3: 1. The date was collected for March 31st and April 2nd, 2020. Thats perhaps a very small time for a study of this magnitude to be conducted

2. the word SWB ?? is present in the abstract, what does it stand for ?

3. The sentence "However, it is unclear whether people in quarantine are able to engage in such frequency and regularity of exercise and if this practice is enough to provide SWB." Was it also measured/investigated in the present research.

4. From the methodology I feel three Instruments were administered to the same population , Do the authors think RANOVA/MANOVA should had been used to control type 2 errors ?

5. Another question that arises is how the author controlled the Intensity or administered the intensity for exercises for the participants during the pandemic , as specific intensity will produce specific results

6. Refer Can moderate intensity aerobic exercise be an effective and valuable therapy in preventing and controlling the pandemic of COVID-19?Med Hypotheses. 2020 Oct; 143: 109854. and Physical exercise as a tool to help the immune system against COVID-19: an integrative review of the current literature, Clinical and Experimental Medicine (2020). Should be included and discussed in the article.

7. Limitations if any

8. I think there is a need to revise the need or gap for the study which needs to be more emphatic so does the conclusion.

6. PLOS authors have the option to publish the peer review history of their article (what does this mean?). If published, this will include your full peer review and any attached files.

Reviewer #1: No

Reviewer #2: No

Reviewer #3: **Yes: **DR SNEHIL DIXIT

---

## [Author Response · Author response to Decision Letter 0]

19 May 2021

1. Is the manuscript technically sound, and do the data support the conclusions?

AUTHOR'S ANSWERS: We reorganized the conclusions accordingly.

2. Has the statistical analysis been performed appropriately and rigorously?

AUTHOR'S ANSWERS: The majority of reviewers considered the statistical analysis adequate.

Reviewer #1: The manuscript covers a very interesting topic and the approach used by the authors is quite robust. However, the way it is written and structured makes it hard for the reader to follow. In addition, some information needs to be better explained by the authors. Detailed comments:

- The manuscript would be both more compelling and useful to a broad readership if the authors moved beyond providing a descriptive narrative of the data analyzed in an assembled way and be more forthright and clear in presenting the instruments and the results (maybe using tables).

- The number of acronyms should be considerably reduced, as it makes the text hard to follow, particularly in the methods section. By the way, the acronym of Positive Affect should be PA and not AP.

- It is not clear if all participants answered all the questionnaires’ items.

- The way results are presented is rather confusing: subtopics should be added to direct the reader.

- The correlation between the sociodemographic data collected and the instruments should be clearer and better explored.

- A "Limitations" section at the end of the "Discussion" and as well as a “Conclusion” section should be included. In the “Conclusion” section, policy implications of the results achieved should be added: conclusion section was inclued.

AUTHOR'S ANSWERS: We changed the presentation of the manuscript according to suggestions. In data analysis, we modified the instrument description and provided a better look at results to make them clearer. The number of acronyms was reduced and the acronym of Positive Affect that should be PA was modified. We made clear that all participants answered all items. We added subtopics in the results, additionally. We also included policy implications of the results in the discussion.

Reviewer #2: The manuscript is of general interest and conducted with a rigorous research approach. I am satisfied with the work done by the authors. I have only the following minor comments to rise.

ABSTRACT

- The authors reported the acronym SWB without having defined it first

- Please provide age of the sample and gender distribution

- Why did the authors only report strength training results? Authors should better report the main findings found.

 INTRODUCTION

- The introduction is too long and dispersed. Authors should streamline it and make it converge on the topics related to the study (for example, the description of the symptoms of COVID-19 or the difference between symptomatic and asymptomatic is not necessary for the purpose of the study)

- Some sentences are not supported by bibliographic sources (for example the following sentence: “However, the same physical distance, which protects COVID-19 from spreading, may have an impact on mental health and the well-being of the population.” For the latter statement I suggest considering the following references:

1) Giustino, V. et al. Physical Activity Levels and Related Energy Expenditure during COVID-19 Quarantine among the Sicilian Active Population: A Cross-Sectional Online Survey Study. Sustainability 2020, 12, 4356.

2) López-Bueno, R. et al. Health-Related Behaviors Among School-Aged Children and Adolescents During the Spanish Covid-19 Confinement. Frontiers in paediatrics 2020, 8, 573.

DISCUSSION

- In the discussion, same sentences concerning the PAWEC should be reported in the methods section (for example the following sentence: “the scale as a whole showed Cronbach’s alpha above 0.70, which was considered enough to assume good internal consistency. The instrument, then, proved to be reliable and well structured”)

- Before describing the results found, the authors should report what their hypothesis was.

- The authors failed to report strength and limitations of the study. Among the latter, the self-reported based instrument should be mentioned.

AUTHOR'S ANSWERS: We adopted all suggestions in the abstract and Introduction including the new references. Regarding the PAWEC, information about the scale was presented in the discussion because it was developed for this study (which is highlighted in the Method section), thus, we decided to further discuss its psychometric properties beyond the main goal of the manuscript. It is pivotal to mention that three-factor structure of PAWEC was found due to this study, with no previous hypothesis regarding dimensional organization.

Reviewer #3: 

1. The date was collected for March 31st and April 2nd, 2020. Thats perhaps a very small time for a study of this magnitude to be conducted.

2. the word SWB ?? is present in the abstract, what does it stand for ?

3. The sentence "However, it is unclear whether people in quarantine are able to engage in such frequency and regularity of exercise and if this practice is enough to provide SWB." Was it also measured/investigated in the present research

4. From the methodology I feel three Instruments were administered to the same population, Do the authors think RANOVA/MANOVA should had been used to control type 2 errors?

5. Another question that arises is how the author controlled the Intensity or administered the intensity for exercises for the participants during the pandemic , as specific intensity will produce specific results.

6. Refer Can moderate intensity aerobic exercise be an effective and valuable therapy in preventing and controlling the pandemic of COVID-19?Med Hypotheses. 2020 Oct; 143: 109854. and Physical exercise as a tool to help the immune system against COVID-19: an integrative review of the current literature, Clinical and Experimental Medicine (2020). Should be included and discussed in the article.

7. Limitations if any.

8. I think there is a need to revise the need or gap for the study which needs to be more emphatic so does the conclusion.

AUTHOR'S ANSWERS:

1. We appreciate this suggestion, the manuscript was modified accordingly.

2. We mention this issue in the end of the manuscript.

3. Although we understand that our time frame was small, this study was conducted in the beginning of the pandemic looking at a critical moment, before participants got used to quarentine. Additionally, social isolation began to fade after April 2020 in Brazil which would impair the mais focus of the study. 

4. Indeed we found that frequency of exercise before isolation is associated to higher SWB during isolation.

5. We believe that providing the power of the test was enough to present to readers the amount of type 2 errors, since statistical power is 1 – β. The main problem of using MANOVA would be the distinct nature of measures employed in our research and we would, by doing that, increase the amount of type 1 error, which was also verified by the effect-size. Though, we thank the reviewer for the suggestion.

6. Indeed we measured frequency of exercise, not the intensity, which was included in the Limitations section.

7. We added a Limitations subsection in the Dicussion section.

8. We modified the text of the manuscript accordingly.

---

## [Decision Letter · Decision Letter 1]

28 Jul 2021

PONE-D-20-25501R1

Effects of physical activity and exercise on well-being in the context of the Covid-19 pandemic

PLOS ONE

Dear Dr. Filgueiras,

Thank you for submitting your manuscript to PLOS ONE. After careful consideration, we feel that it has merit but does not fully meet PLOS ONE’s publication criteria as it currently stands. Therefore, we invite you to submit a revised version of the manuscript that addresses the points raised during the review process.

reviewers pointed out that the new version of the manuscript is of better quality, but they still highlight points that require the authors' attention, for example:

- language / English must be revised

-bibliographic references

- study limitations 

We look forward to receiving your revised manuscript.

Kind regards,

Flávia L. Osório, PhD

Academic Editor

PLOS ONE

Journal Requirements:

Reviewers' comments:

Reviewer's Responses to Questions

**Comments to the Author**

1. If the authors have adequately addressed your comments raised in a previous round of review and you feel that this manuscript is now acceptable for publication, you may indicate that here to bypass the “Comments to the Author” section, enter your conflict of interest statement in the “Confidential to Editor” section, and submit your "Accept" recommendation.

Reviewer #1: (No Response)

Reviewer #2: (No Response)

2. Is the manuscript technically sound, and do the data support the conclusions?

Reviewer #1: Yes

Reviewer #2: Yes

3. Has the statistical analysis been performed appropriately and rigorously? 

Reviewer #1: Yes

Reviewer #2: Yes

4. Have the authors made all data underlying the findings in their manuscript fully available?

Reviewer #1: Yes

Reviewer #2: Yes

5. Is the manuscript presented in an intelligible fashion and written in standard English?

Reviewer #1: No

Reviewer #2: Yes

6. Review Comments to the Author

Reviewer #1: The authors have improved the manuscript taking into account most of my previous comments. However, there are important issues that need to be considered before publication. First, the English in the present manuscript is not of publication quality and requires major improvement. There are several mistakes and sentences not completed, which make the manuscript difficult to read. I therefore suggest that the authors get editing help from someone with full professional proficiency in English. In addition, some references are numbered in the text and others are not. Some references mentioned in the text are also missing in the bibliography section. Finally, the conclusion needs to be improved with the limitations of the study (the authors only introduced an unfinished sentenced on this) and possible paths for future research.

Reviewer #2: Overall, it is a piece of general interest and rigorously carried out. The methodological approach and statistical analysis are appropriate, and the conclusions represent a reasonable extension of the results. I suggest minor revisions before the acceptation.

The abstract should contain information on the study design.

Some references are missing in the introduction. For example: “On March 11th, 2020, Covid-19 was characterized by … as a pandemic” ; “However, the same physical distance, … and the well-being of the population” ; “In a 2004 study of 129 Canadian people … were observed” ; etc.

In the statement “However, it is unclear whether people in quarantine are able to engage in such frequency and regularity of exercise and if this practice is enough to provide SWB”, please consider the following reference about the practice of physical activity during the Covid-19 pandemic: “Di Stefano V, et al. Significant reduction of physical activity in patients with neuromuscular disease during COVID-19 pandemic: the long-term consequences of quarantine. J Neurol. 2021; 268(1):20-26. doi: 10.1007/s00415-020-10064-6.”

In the procedures authors should define the study design.

The authors stated that "Volunteers were recruited via authors’ social media and messaging smartphone apps". Since methods should be replicable, sample recruitment information should be better reported.

In the instruments paragraph, authors stated that “Three instruments were used online and sent in a single form of Google Docs”. However, it is not clear whether the link received by the participants to answer the questionnaire (as described in the procedures paragraph) was the Google Docs form or if, instead, only the answers were sent to Google Docs.

7. PLOS authors have the option to publish the peer review history of their article (what does this mean?). If published, this will include your full peer review and any attached files.

Reviewer #1: No

Reviewer #2: No

---

## [Author Response · Author response to Decision Letter 1]

19 Oct 2021

Dear Editor,

In behalf of the other authors, I can say that we appreciate the effort of both Editor and Reviewers in making this manuscript better and ready for publication at Plos One. We tried to tap into the last requirements that were asked. We made a full language and format review with a native English speaker in order to make it suitable for publication. We thank the opportunity and look forward to hearing from Plos One. Below, the list of modifications.

1) Citations were completely reviewed and reorganized according to Plos One format (Vancouver).

2) Language was reviewed by a native English speaker.

3) To improve clarification, the term disposed was replaced by energetic in Table 1 (pg. 14)

4) Last paragraph of results (pg. 16) was completely rewritten for better understanding of the reader.

5) Last phrase of the first paragraph of Discussion (pp. 16-17) was rewritten due to language issues.

6) First paragraph of the first subsection of the Discussion (pg. 18) was changed to improve understanding.

7) In-text citations were reviewed to be adequate to Vancouver format.

---

## [Decision Letter · Decision Letter 2]

11 Nov 2021

Effects of physical activity and exercise on well-being in the context of the Covid-19 pandemic

PONE-D-20-25501R2

Dear Dr. Filgueiras,

We’re pleased to inform you that your manuscript has been judged scientifically suitable for publication and will be formally accepted for publication once it meets all outstanding technical requirements.

Kind regards,

Flávia L. Osório, PhD

Academic Editor

PLOS ONE

Additional Editor Comments (optional):

Reviewers' comments:

Reviewer's Responses to Questions

**Comments to the Author**

1. If the authors have adequately addressed your comments raised in a previous round of review and you feel that this manuscript is now acceptable for publication, you may indicate that here to bypass the “Comments to the Author” section, enter your conflict of interest statement in the “Confidential to Editor” section, and submit your "Accept" recommendation.

Reviewer #1: All comments have been addressed

Reviewer #2: (No Response)

2. Is the manuscript technically sound, and do the data support the conclusions?

Reviewer #1: Yes

Reviewer #2: (No Response)

3. Has the statistical analysis been performed appropriately and rigorously? 

Reviewer #1: Yes

Reviewer #2: (No Response)

4. Have the authors made all data underlying the findings in their manuscript fully available?

Reviewer #1: Yes

Reviewer #2: Yes

5. Is the manuscript presented in an intelligible fashion and written in standard English?

Reviewer #1: Yes

Reviewer #2: Yes

6. Review Comments to the Author

Reviewer #1: The authors have addressed most of my previous remarks. In my opinion, the paper is now ready to be published.

Reviewer #2: In my opinion the manuscript was improved according to Reviewers' indications. For these reasons it can be accepted for pubblication in the current form.

Best Regards

7. PLOS authors have the option to publish the peer review history of their article (what does this mean?). If published, this will include your full peer review and any attached files.

Reviewer #1: No

Reviewer #2: No

---

## [Editor Report · Acceptance letter]

23 Dec 2021

PONE-D-20-25501R2 

Effects of physical activity and exercise on well-being in the context of the Covid-19 pandemic 

Dear Dr. Filgueiras:

I'm pleased to inform you that your manuscript has been deemed suitable for publication in PLOS ONE. Congratulations! Your manuscript is now with our production department. 

Kind regards, 

on behalf of

Dr. Flávia L. Osório 

Academic Editor

PLOS ONE